# Hybrid Biocomposites Based on Poly(Lactic Acid) and Silica Aerogel for Food Packaging Applications

**DOI:** 10.3390/ma13214910

**Published:** 2020-10-31

**Authors:** Alejandro Aragón-Gutierrez, Marina P. Arrieta, Mar López-González, Marta Fernández-García, Daniel López

**Affiliations:** 1Instituto de Ciencia y Tecnología de Polímeros, ICTP-CSIC, Juan de la Cierva 3, 28006 Madrid, Spain; alejandro.aragon.gutierrez@gmail.com (A.A.-G.); mar@ictp.csic.es (M.L.-G.); martafg@ictp.csic.es (M.F.-G.); 2Departamento de Ingeniería Química y del Medio Ambiente, Escuela Politécnica Superior de Ingenieros Industriales, Universidad Politécnica de Madrid (ETSII-UPM), Calle José Gutiérrez Abascal 2, 28006 Madrid, Spain; 3Grupo de Investigación: Polímeros, Caracterización y Aplicaciones (POLCA), 28006 Madrid, Spain; 4Interdisciplinary Platform for Sustainable Plastics towards a Circular Economy-Spanish National Council (SusPlast-CSIC), 28006 Madrid, Spain

**Keywords:** biocomposites, biodegradable polymers, plasticizers, food packaging, poly (lactic acid), citrate esters, silica aerogel

## Abstract

Bionanocomposites based on poly (lactic acid) (PLA) and silica aerogel (SiA) were developed by means of melt extrusion process. PLA-SiA composite films were plasticized with 15 wt.% of acetyl (tributyl citrate) (ATBC) to facilitate the PLA processability as well as to attain flexible polymeric formulations for films for food packaging purposes. Meanwhile, SiA was added in four different proportions (0.5, 1, 3 and 5 wt.%) to evaluate the ability of SiA to improve the thermal, mechanical, and barrier performance of the bionanocomposites. The mechanical performance, thermal stability as well as the barrier properties against different gases (carbon dioxide, nitrogen, and oxygen) of the bionanocomposites were evaluated. It was observed that the addition of 3 wt.% of SiA to the plasticized PLA-ATBC matrix showed simultaneously an improvement on the thermal stability as well as the mechanical and barrier performance of films. Finally, PLA-SiA film formulations were disintegrated in compost at the lab-scale level. The combination of ATBC and SiA sped up the disintegration of PLA matrix. Thus, the bionanocomposites produced here show great potential as sustainable polymeric formulations with interest in the food packaging sector.

## 1. Introduction

As a result of environmental awareness, the use of the petrochemical resources to produce short term plastics products as well as the disposal of large amounts of waste from daily use polymers, such as food packaging, is among one of the foremost concerns in current times. Therefore, the use of both biobased and biodegradable polymers for food packaging applications emerges as a promising solution to this rapidly rising waste problem with the main objective to reduce the consumption of non-renewable resources and prevent the accumulation of plastics wastes [1,2]. Among all biobased and biodegradable thermoplastic polymers currently used in the packaging field (i.e., poly (lactic acid) (PLA), poly (hydroxyalcanoates) (PHAs), thermoplastic starch (TPS), and poly (butylene adipate co-terephthalate) (PBAT)), PLA is the most widely used and investigated for short-term applications [2,3,4]. In this sense, PLA presents a series of promising properties interesting for film applications intended for agricultural mulch films or food packaging, such as high transparency, excellent printability, availability in the market, economically competitive, ease of processing by the already existing technologies in the plastic processing industry (i.e., melt-blending, extrusion, injection molding and film forming), and it presents inherent environmentally friendly characteristics (i.e., it is biobased and biodegradable) [2,3,5]. In this sense, it is synthesized by the ring opening polymerization of lactide (LA) which in turn is coming from the fermentation of renewable agricultural crops (corn, sugar-cane and other polysaccharides). Moreover, it is compostable showing a high rate of disintegration disappearing completely in less than one month [6,7,8,9]. Thus, PLA based products can be wasted along with organic waste instead of being discarded with the rest of plastic materials [10]. This is why PLA represents a sustainable alternative to replace non-compostable conventional polymers derived from petroleum. However, PLA presents some drawbacks with respect to their petrochemical counterparts that restrict its application as flexible film for food packaging such as its low thermal stability, limited stretchability, poor barrier properties, low crystallization rate, and degree [3].

Blending approaches are the most widely used to improve PLA properties for industrial applications. Therefore, the poor ductility of PLA is usually enhanced with the addition of a plasticizer, although it has been reported that the addition of a plasticizer to the PLA matrix increases gas permeability, which may be detrimental for food packaging applications [11,12,13]. Several monomers and oligomers have been investigated as plasticizers for PLA, but citrate esters are probably the most commonly used and investigated since they are accepted for food contact, available at the market and economical competitive [14,15,16]. Among all citrate esters used as plasticizers in the food packaging industry, triethyl citrate (TEC), tributyl citrate (TBC), acetyl triethyl citrate (ATEC), and acetyl tributyl citrate (ATBC) are noteworthy [16,17,18]. In this sense, ATBC has shown good thermal processability as well as effective plasticization effect, and therefore is one of the most recommended and widely used plasticizers for PLA and other biopolyesters [12,14,19]. To obtain the required flexibility for food packaging applications, citrate esters should be added in amounts between 10 and 20 wt.% [7,12,14,20,21]. Additionally, because of its low toxicity, citrate esters are accepted for food contact [22]. However, PLA plasticization will further reduce the well-known PLA poor barrier performance [21,23]. Thus, different strategies have been used to improve plasticized PLA barrier properties including multilayer approaches, surface film coating, co-extrusion with other biopolymers [21,23], and increasing its crystallization degree by blending, co-polimerization [24,25] as well as the development of composites and/or nanocomposites [12,14,26,27].

Composites have been positioned as a promising option in the provision of excellent barrier properties for food packaging applications. Inorganic materials such as silica exhibits excellent thermal stability and higher hardness [28]. In this sense, silica based particles are low-price particles widely used at industrial level since the incorporation of well dispersed silica particles into the polymeric matrix was proven to be an effective way to improve the thermomechanical performance of biopolymers [28,29]. Aerogels are ultra-light weight open-celled mesoporous materials whose structure is composed of an interconnected network of channels made of thin ligaments [28]. For the development of the present work, silica aerogels (SiA) have been chosen as reinforcement for the plasticized polymer matrix. SiA are three-dimensional highly porous materials with nanoporous networks, high specific surface and extremely low density, while polymer-silica aerogel based composites tend to exhibit enhanced mechanical and thermal properties [30,31,32]. SiA reinforced materials has also been proposed for food contact packaging applications. For instance, Chen et al. developed films based on poly vinyl alcohol (PVA) incorporated with SiA for food packaging with improved thermal insulation and oxygen barrier performance [32]. Ventaka Prasad et al. have used SiA particles to improve the mechanical properties of PLA/sisal composites and they conclude that the improvement of the interfacial adhesion between the polymeric matrix and sisal fibers due to the SiA particles presence increased the tensile and flexural strength of the final materials [28].

Thus, considering that among the main drawbacks of PLA to be used in food packaging are its rigidity, its low gas barrier properties, and its low flexibility, the main objective of the present work is the development of hybrid biocomposites based on plasticized PLA with 15 wt.% of a frequently used PLA plasticizer, such as ATBC, and further reinforced with different amounts of silica aerogels (SiA) particles (0.5, 1, 3, and 5 wt.%) to obtain biocomposite films with enhanced barrier, thermal as well as mechanical properties. Additionally, the disintegration behavior under composting medium at laboratory scale level of the SiA nanoreinforced plasticized PLA formulations was studied to corroborate their compostable character.

## 2. Materials and Methods

### 2.1. Materials

Poly (lactic acid) was kindly provided by ERCROS S.A. (ErcrosBio^®^ LL703 density 1.25 g cm^−3^) (Barcelona, Spain) M_n_ = 23,250 Da [7]. Acetyl tri-n-butyl citrate (ATBC, M = 402 g mol^−1^, 98% purity) was purchased from Sigma-Aldrich (Madrid, Spain). Silica aerogel (SiA) powder was kindly provided by JIOS Aerogel (Seoul, Korea) (particle size 1–20 μm, pore diameter 20 nm and surface area 600–800 m/g^2^).

### 2.2. Film Samples Preparation

PLA and plasticized PLA biocomposites were prepared in a twin conical co-rotating screws (Thermo Scientific MiniLab Haake Rheomex CTW5) (Waltham, MA, USA) with a capacity of 7 cm^3^. PLA undergoes hydrolytic degradation, which is influenced by ambient moisture and temperature [33], in order to avoid the presence of moisture PLA pellets were previously dried overnight at 80 °C [9,33,34,35]. The same procedure was applied to SiA, while ATBC was dried at 80 °C during 2 h [12]. The appropriate amounts of PLA and SiA were firstly processed in the Minilab mixer (Waltham, MA, USA) operating at a rotation speed of 100 rpm and at 190 °C. Once PLA had reached the melt state (after 2 min), ATBC was added in 15 wt.% to avoid ATBC thermal degradation, following an already optimized process [21]. Therefore, the total time of biocomposite formulations mixing was 3 min. Films were then processed from the obtained extruded blends by compression molding process at 200 °C in a hot press (Collin P-200-P) (GmbH, Ebersberg, Germany). Each pelletized blend formulation (1 g) was kept between the hot plates firstly at atmospheric pressure and at 200 °C until reaching the melt state (one minute). Subsequently, the hot press was submitted to an already optimized pressure cycles for PLA-ATBC based formulations, 1 MPa for one minute, 5 MPa for one minute, and 5 MPa for two minutes [7]. Six formulations were obtained and were named as shown in Table 1.

### 2.3. Thermal Characterization

#### 2.3.1. Differential Scanning Calorimetry (DSC)

DSC was used to determine glass transition temperature (T_g_), cold crystallization temperature (T_cc_), melting temperature (T_m_) and degree of crystallinity (χ_c_). Experiments were conducted in a DSC calorimeter Q-2000 (New Castle, DE, USA). The DSC runs were performed in a nitrogen atmosphere. Samples (around 3.5 mg) were subjected to a cycle program consisting in a first heating run from 25 °C to 200 °C (rate = 10 °C min^−1^) followed by a cooling process from 200 °C to −50 °C and finally a second heating up to 200 °C at a rate of 10 °C min^−1^. The DSC thermal parameters: glass transition temperature (T_g_), the melting temperature (T_m_), and cold crystallization temperature (T_cc_), were obtained from the first DSC heating scan. The degree of crystallinity (χ_c_) of PLA, PLA-ATBC and its plasticized biocomposites was calculated by using Equation (1):(1)$$\chi_{c}=100\times \left[\frac{{\Delta H}_{m}-{\Delta H}_{\mathrm{cc}}}{{\Delta H}_{m}^{c}}\right]\;\times \;\frac{1}{W_{\mathrm{PLA}}}$$
where ∆H_m_ and ∆H_cc_ are the melting enthalpy and the cold crystallization enthalpy of the sample, respectively. ${\Delta H}_{m}^{c}$ is the melting heat of purely crystalline PLA taken as 93 J/g [36].

#### 2.3.2. Thermogravimetric Analysis (TGA)

Thermogravimetric dynamic measurements were run in a thermogravimetric analyzer (TGA Q-500) (TA Instruments, New Castle, DE, USA) at a constant heating rate of 10 °C/min. Temperature was raised from 30 to 700 °C under nitrogen atmosphere. The onset degradation temperature (T_0_) was calculated at 5% weight loss [7]; meanwhile, the first derivative of the TGA curves (DTG) was used to calculate the maximum degradation rate temperatures (T_max_) for each stage.

### 2.4. Mechanical Characterization

Mechanical properties of the samples were obtained by using a universal testing system (3366 Instron dynamometer) (INSTRON, Norwood, MA, USA), using a 100 N cell and 10 mm min^−1^ of speed. Dimensions of the halterio type probes, were 35 mm length, 2 mm width and a thickness at around 0.2 mm. Separation between the clamps was 20 mm. Results to obtain the Young modulus (E), tensile strength (TS), and the average percentage elongation at break (ε_B%_) were calculated from the obtained stress–strain curves as the average of at least five measurements. Significance in the mechanical properties differences were statistically determined by means of one-way analysis of variance (ANOVA) with OriginPro 8.6 software (OriginLab corporation Northampton, MA, USA). Means comparison was performed employing a Tukey’s test, at 95% of the confidence level, in order to identify which formulations were significantly different from other formulations.

### 2.5. Scanning Electron Microscopy (SEM)

SEM micrographs of the crio-fractured cross-section of the films were obtained by means of a Philips XL30 SEM (Phillips, Eindhoven, The Netherlands). PLA and plasticized PLA formulations were previously coated under vacuum conditions with a palladium/gold layer previous to their SEM observation with the main objective to enhance their electrical conductivity.

### 2.6. Permeability to Gases

Permeation measurements of O_2_, CO_2_, and N_2_ through the films were carried out in a lab-made permeator, described elsewhere by Tiemblo et al. [37]. In brief, biocomposite films were placed in the center of a cell to separate the upstream chamber (the high pressure chamber) from the downstream chamber (low-pressure chamber). The system was thermostatically controlled in a water bath at 30 °C. Then, the permeation device was brought to a high vacuum with an Alcatel turbomolecular pump. The vacuum was maintained overnight to remove the last traces of gas and solvent in the membrane until low pressure was reached, which is about 10^−6^ mbar. Subsequently, the measurements were performed, and thus the gas in a reservoir at the pressure of interest was allowed to flow into the downstream chamber, and it was further monitored as a function of time with an MKS 628B Baratron (MKS Instruments Inc., Andover, MA, USA), which is an absolute pressure transducer which worked in the 0–10 mbar pressure range, connected to a computer. A Gometric transducer of 0–10 bar range measured the gas pressures in the upstream chamber. The permeability (*P*) and diffusion (*D*) coefficients were obtained from the plotted curves measuring the pressure increases in the downstream chamber at intervals of 1 s. Before each measurement, the inlet of air was measured as a function of time and it was then subtracted from the obtained curves [38]. Thus, the pressure of permeant versus time was plotted. Biocomposite films with thickness between 100 to 200 μm were used and three independent experiments were carried out for each gas. The volume *V* of the downstream semicell is given in cm^3^, the thickness (*L*) in cm, the diffusion area (*A*) of the membrane is in cm^2^, and the pressure in cm Hg, and thus the permeability coefficient (*P*) of each gas (O_2_, CO_2_ and N_2_) in the membranes is reported in barrer (1 barrer = 10^−10^ cm^3^(STP)cm/(cm^2^ s cmHg)) following Equation (2):(2)$$P=\;\frac{273}{76}\left(\frac{VL}{ATp_{0}}\right)\left(\frac{dp\left(t\right)}{dt}\right),$$
where *p*_0_ is the upstream gas pressure, *p* is downstream gas pressure, and *T* is the absolute temperature. The *p* against *t* isotherms present a transitory state followed by a straight line (*t*→ ∞), which corresponds to the conditions in the steady-state. The intersection of this straight line with the abscissa axis of the plot represents the time lag *θ*. This time lag *θ* is related by means of the Equation (3) [38] to the apparent gas diffusion coefficient (*D*) in the:(3)$$D=\;\frac{L^{2}}{6\theta }$$

The diffusion coefficient (*D*) is given in cm^2^ s^−1^. The apparent solubility coefficient (*S*) is given in (cm^3^(STP)/(cm^3^ cm Hg)), following Equation (4):(4)$$S=\;\frac{P}{D}$$

### 2.7. Disintegrability under Composting Conditions

Disintegration under composting conditions at a laboratory scale level was performed based on the UNE-EN ISO-20200 standard [39]. The solid synthetic waste (SSW) was elaborated by mixing 10% of compost, 30% rabbit food, 40% sawdust, 10% starch, 5% sugar, 1% urea, and 4% corn oil. Water was then added to the SSW in 45:55 SSW:water proportions. All formulations were cut in 1.5 × 1.5 cm^2^ and buried inside a polyester mesh to permit taking away the disintegrated film materials from the reactors containing the compost soil, but at the same time allowing the access of moisture and furthermore the access of microorganisms [40]. The materials were further subjected to aerobic disintegration conditions in composting containers at 58 °C in an oven for a period of 35 days. Water was added periodically to the composting containers maintain the required humidity. Each film formulation was recovered from the composting containers at 1, 4, 10, 21, 28, and 35 days. Each recovered sample was cleaned using distilled water and then dried at 40 °C in an oven for around 24 h and further re-weighted. The disintegration degree of each formulation was calculated by normalizing the sample weight at each disintegration time to the initial weight. Additionally, photographs were taken to all recovered disintegrated films from the composting containers in order to qualitatively follow the disintegration process. The degree of disintegration was calculated following Equation (5):(5)$$GD=\frac{m_{i}-m_{r}}{m_{i}}\times 100$$
where *GD* is the degree of disintegration, *m_i_* is the initial dry mass of the tested formulation, and *m_r_* is the final dry mass of the recovered sample.

## 3. Results

The processing technology used for the development of SiA reinforced plasticized PLA formulations allowed us to obtain transparent films with thickness ranging between 100 µm and 200 µm. The visual appearance of the PLA-ATBC-SiA3% film formulation is showed in Figure 1a as an example. The films showed good transparency after the addition of the plasticizer (ATBC) in agreement with already reported works [7,12,14,21], even after the addition of silica aerogel suggesting good dispersion of this filler into the plasticized PLA matrix. These findings were confirmed by SEM. Thus, morphological characteristics of the cross cryo-fracture sections of film samples were evaluated by SEM and the images are shown in Figure 1. As expected, PLA presented a smooth and uniform surface where a rigid fracture typical of amorphous polymer [9] could be observed (Figure 1b), while plasticized samples presented plastic deformation, showing more ductile fracture patterns [12] (Figure 1c–h), in accordance with previous works [12,21,41]. In the case of silica aerogel based composites, the fracture surface resulted in more roughness (Figure 1d–h) as SiA concentration increased in the formulation. Small white spots on the cross-section of the plasticized PLA-ATBC-SiA films can be observed, and they became denser with increasing SiA content. Similar findings have been observed in PVA based films loaded with SiA [32]. At higher magnifications, it could be observed that silica aerogel particles appeared well dispersed as crystalline aggregates in the polymeric matrix, as can be seen in PLA-ATBC-SiA1% (Figure 1h). This fact influenced that the films were still transparent after the addition of the silica aerogel particles (see Figure 1a). In fact, the well dispersion of silica aerogel particles suggests that no phase separation had taken place during the extrusion process.

The thermal stability of all formulations was studied by thermogravimetric analysis. The main thermal parameters obtained from TGA and DTG curves are summarized in Table 2, while DTG curves are shown in Figure 2a. Neat PLA film decomposed in a single degradation step process with an onset degradation temperature (T_0_) of 322 °C, while the maximum degradation rate temperature (T_max_) was centered at 360 °C. The presence of ATBC plasticizer decreased the onset degradation temperature (T_0_) of the polymeric material. A slightly growth of the residual weight percentage at the end of the TGA analysis was evident in those formulation with a higher amount of SiA, 3 wt.% and 5 wt.%.

The DTG curves of Figure 2a showed that PLA-ATBC showed a degradation peak prior to the main degradation of the polymeric matrix, which has been already observed ascribed with ATBC plasticizer vaporization [21]. Conversely, the addition of silica aerogel increases the onset degradation temperature (T_0_) with respect to the plasticized PLA-ATBC films suggestive of some stabilization of the PLA-ATBC continuous phase. Regarding the T_max_ values corresponding to the PLA-ATBC and PLA-ATBC reinforced with aerogel, no significant changes were not observed in either of the two degradation stages, that of plasticizer and that of the polymeric matrix. In conclusion, reinforcing plasticized PLA with silica aerogel resulted in little changes in their thermal stability. Nevertheless, it should be highlighted that not significant modifications on the TGA thermal parameters were observed at the processing as well as the intended use temperatures and thus confirming the good thermal stability of the developed biocomposites.

Figure 2 shows the DSC thermograms of all samples during the first heating scan (Figure 2b), the cooling scan (Figure 2c), and the second heating scan (Figure 2d). Meanwhile, Table 3 summarizes the corresponding values of the main thermal parameters. As expected, the addition of ATBC had a clear effect on the plasticization of the PLA polymeric matrix, since it induced a decrease in the T_g_ in all plasticized films. This reduction in the T_g_ has been ascribed to the ability of citrate plasticizers to increase the mobility of the PLA polymer chains, caused by an increase in the free volume between them [14,21,41]. Although, from the obtained DSC curves in blend formulations, the two parallel baselines, before and after the T_g_, are not well defined, it is evident that the typical T_g_ of PLA could not be seen at 58 °C, suggesting that, in PLA, reinforced films further decreased the T_g_, with respect to the PLA-ATBC sample. The midpoint between intersections can be estimated and these values are summarized in Table 3. The disappearance of PLA T_g_ at 59 °C confirms the good interaction of the silica aerogel microparticles with the polymeric matrix as well as with the plasticizer in the blend.

The cold crystallization in neat PLA suggests that the processing conditions did not induce the crystallization of PLA. Nevertheless, in the case of plasticized systems, it could be observed a clear decrease not only in the enthalpy, but also in the cold crystallization temperature with respect to the neat PLA, suggesting that ATBC favors PLA crystallization during processing. Indeed, the degree of crystallinity was higher for all plasticized formulations (Table 3). It has been observed that ATBC plasticizer interacts well with PLA matrix, and thus enhancing the PLA crystallization rate [7,16]. Although no significant variations in the cold crystallization temperature between the plasticized PLA-ATBC films and composites reinforced with silica aerogel were observed, it seems that the combination of silica aerogel with ATBC plasticizer in the PLA based blend formulations lead to a slightly increase of the T_cc_ value. This synergic effect on the crystallization of PLA plasticized with ATBC and a potential nucleating agent has already been reported in PLA-ATBC reinforced with talc [20] and nanocellulose [12].

It is remarkable that neat PLA presents a small exothermic peak right before melting, which disappears with the presence of the plasticizer. This peak has been related with restrictions in the polymer chains due to the transition of the disorder to order PLA crystallites [42]—thus suggesting that a great part of the polymeric system was in an amorphous state as a consequence of the fast DSC cooling rate applied after the compression molding process [7]. As expected, ATBC reduced the melting temperature of PLA confirming again the good interaction between the ATBC plasticizer and the polymeric matrix, in accordance with previous works [7,21,41]. The melting temperatures of the composites resulted in being almost invariant with respect to the PLA-ATBC film.

In plasticized formulations, the addition of ATBC plasticizer to the PLA polymer matrix induced an increase in the degree of crystallinity, related to the fact that the ATBC chains can interact with the polymeric chains of PLA, which result in increased polymeric chains mobility, which influences the nucleation and thus promotes a better packaging of segments [23]. However, the degree of crystallinity was reduced increasing the silica aerogel amount in the formulation, suggesting that higher silica aerogel contents restrict the polymeric chains mobility in the system, which is required for its regular packing into crystal lattices, because of enlarged interaction sites between PLA matrix and the inorganic silica phase, ultimately retarding the crystallization.

PLA is well known for its slow nucleation and crystallization rates. In the present work, the cooling DSC conditions conducted at slow rate allowed PLA crystallization (Figure 2c) in all formulations, while T_g_ values could not be observed during the cooling probably due to the low amount of sample used. Thus, all samples presented higher crystallinity degrees during the second DSC heating (Figure 2d and Table 3) than in the first DSC heating (Figure 2b and Table 3). The second DSC heating allowed better comparison between samples as a consequence of the elimination of the PLA-based formulations thermal history (Figure 2d). Neat PLA was the only film that showed the cold crystallization in the second DSC heating. This result shows that PLA was not capable to crystallize completely, even at the low cooling rate used during DSC cooling scan. During the second DSC heating (Figure 2d), all the plasticized formulations did not show the cold crystallization and only the melting temperature is clearly observed, showing complete crystallized materials, highlighting the ability of ATBC plasticizer to crystallize PLA during the cooling.

The mechanical behavior of the different plasticized PLA-based materials was assessed and the tensile test results are reported in Figure 3. The tensile test results for neat PLA film are also reported for comparison. Figure 3a shows the obtained results of the Young’s Modulus (*E*) for all tested samples. The high Young’s modulus values corresponding to the neat PLA are symptomatic of the high stiffness and brittleness that typically present the neat polymer, with elastic modulus values approaching 2.5 GPa. As expected, the addition of ATBC caused a significant (*p* < 0.05) reduction in the Young’s modulus up to 60%, since plasticizers induce ductile fracture. In general, biocomposites presented comparable values of elastic modulus, showing not significant (*p* > 0.05) changes with respect to the plasticized PLA-ATBC film, with the exception of PLA-ATBC-SiA3% which showed some (*p* > 0.05) improvement of the Young’s modulus with respect to PLA-ATBC suggesting a reinforcing effect.

In the same way as for the Young’s modulus, the tensile strength decreased significantly (*p* < 0.05) with the presence of the plasticizer (Figure 3b). The incorporation of silica aerogel slightly reduced the tensile strength of PLA-ATBC film (*p* > 0.05). The presence of ATBC in the plasticized PLA and its biocomposites produced a reduction in the tensile strength up to 55% when compared to neat PLA (*p* < 0.05).

Regarding the elongation at break (Figure 3c), neat PLA did not show an elastic behavior, presenting a value of elongation at break of 4% demonstrating the well-known poor flexibility of PLA. A noticeable significant (*p* < 0.05) improvement in elongation at break values was evidenced for PLA plasticized with ATBC and its biocomposite films, in good accordance with the reduction in the T_g_ values in these formulations as well as with already reported works [7,12,21]. The elongation at break reached values between 140% and 185% in the plasticized samples, as a result of the increase in the polymer chains mobility. The best improvement in the elongation at break of the biocomposites was observed for the biocomposite loaded with the less amount of SiA (PLA-ATBC-SiA0.5%). In this sense, although not statistically significant (*p* > 0.05), it can be observed that the addition of low amounts of SiA, which is 0.5 wt% and 1 wt%, caused a somewhat plasticizing effect.

Similar findings have already been observed in PLA based polyurethanes reinforced with nano hydroxyapatite in which less reinforcing material of 0.5 and 1 wt.% produced a slightly plasticizing effect, while higher amounts of 3 wt.% produced a reinforcing effect [25]. In fact, the less elongation at break value was observed for PLA-ATBC-SiA3% biocomposite formulation (*p* < 0.05), in agreement with the highest Young’s modulus, but still showing enough stretchability for the intended use. In this sense, it should be highlighted that films intended for food packaging applications require high flexibility at room temperature and the elongation at break values obtained here for PLA-ATBC and PLA based biocomposites are comparable in terms of flexibility to frequently used traditional petrol-based and non-biodegradable polymers. One of the most used conventional plastic materials at the industrial level for food packaging applications is low density polyethylene (LDPE), with tensile strength values between 10 and 20 MPa and elongation at break between 80 and 500% [43,44,45]. Therefore, the biocomposite films developed here offer possibilities for their processing at industrial level and their use in the food packaging sector.

Besides containing and protecting packed food from physical damage, food packaging materials should maintain foodstuff quality and extend their shelf-life by controlling the mass transfer processes [46]. Considering that plastic, and particularly bioplastics, are relatively permeable to small molecules [47], the polymer barrier performance in food packaging materials based on biopolymers is of fundamental importance. Table 4 shows the permeability, diffusion, and solubility coefficients values obtained for all formulations by using the time-lag method for CO_2_, N_2_ and O_2_. Meanwhile, for a better comparison of the results of plasticized samples, the diffusivity and permeability parameters for PLA, PLA-ATBC and the biocomposites as a function of the silica aerogel content are represented in Figure 4a,b, respectively.

As it was expected and can be seen in Table 4, the permeability followed the trends P_CO2_ > P_O2_ > P_N2_ in all film samples and the values of apparent solubility coefficients were higher for more condensable gases, so S_CO2_ > S_O2_ > S_N2_. As it was already commented, the addition of ATBC plasticizer increased the free volume leading to an increase in the mobility of the polymer chains. Thus, gases diffused easily through the films and consequently the permeability and diffusion coefficients increased with the addition of ATBC from pristine PLA, in agreement with previously developed PLA-ATBC based materials [12,21]. It could be observed that, for biocomposites with 0.5 wt.% and 1 wt.% of silica aerogel contents, the diffusion coefficients increased for all gases tested. Therefore, in these cases, the permeabilities were improved with the addition of SiA fillers up to 1 wt.%. Nevertheless, higher aerogel contents of 3 wt.% and 5 wt.% in the composite materials provoked a decrease in diffusivity values, which were slightly lower compared to PLA-ATBC formulation. Regarding the permeability, comparable values were obtained between PLA-ATBC film sample and those biocomposites loaded with 3 wt.% and 5 wt.% of silica aerogel.

It is noteworthy that N_2_ has lower diffusivity and permeability compared to CO_2_ and O_2_. In the case of neat PLA film, N_2_ permeability is about 0.06 Barrer, 18 times smaller than CO_2_ and 6 times smaller than O_2_. This N_2_ permeation was too low to be accurately measured in our experimental system. In relation to the apparent solubility coefficient, no significant differences were observed between the plasticized biocomposites film formulations. In all cases, CO_2_ is the gas that presented a higher solubility due to its bigger condensability.

As expected, the increase in the free volume in plasticized systems led to an increment in diffusion coefficient and, thus, in the permeability to gases. By adding low contents of silica aerogel, 0.5 wt.% and 1 wt.%, the diffusivity through the films becomes significantly larger. Considering the good compatibility between the additives and the polymeric matrix (as can be seen in Figure 1), this increase in the diffusion coefficients suggest that a low content of silica aerogel caused a further plasticizing effect, in agreement with the higher elongation at break showed for these biocomposites. It should be also considered that crystalline regions act as an impermeable barrier for the diffusion and adsorption and, as commented before, the addition of silica aerogel decreases the degree of crystallinity, increasing in this way the permeability to gases. On the other hand, for high aerogel contents, 3 wt.% and 5% wt.%, the negative effect of the decrease in the crystallinity degree would have been compensated by a more tortuous pathway for gas diffusion. Thus, in these cases, the result of both effects was the decrease of the diffusion coefficients to values slightly lower than for PLA-ATBC film. The reduction of the diffusion coefficient caused the consequent decrease in the permeability of composites with 3 wt.% and 5 wt.% of aerogel content, reaching permeability values comparable to that PLA-ATBC film. Similarly, Chen et al. need at least 3 wt.% of SiA content to improve PVA barrier performance [32]. This behavior has been related with the good dispersion and good adherence of fillers into the polymeric matrix, which hinders micro-Brownian motions and thus minimizes the formation of non-permanent holes that would promote gas diffusivity through the film [48]. It should be mentioned that it is known that the PLA permeability to gases decreases as temperature decreases [47]. Thus, it is expected that, in foreseeable conditions of the intended use, such as at fridge storage temperature (i.e., 4 °C), the gas permeability values of biocomposites will decrease further.

Biodegradable polymers are able to decompose into simple molecules found in the environment (i.e., carbon dioxide, methane, water, inorganic compounds, or biomass), mediated by the enzymatic action of the microorganisms, in a determined period of time and includes the formation of humus soil that can be further used as soil conditioner [40,49,50]. Thus, finally, the PLA and plasticized PLA formulations were disintegrated under composting conditions performed at a laboratory scale level. The compostable character was qualitatively checked by photographs taken. Figure 5a shows the visual aspect of the recovered film samples at different testing days of exposition to the compost medium. Meanwhile, the degree of disintegration as a function of time was quantitatively followed as it is showed in Figure 5b.

It was observed that all PLA and PLA-ATBC based samples changed their visual appearance during the first week under composting conditions—becoming opaque. The changes of the refraction index of the polymeric materials during composting has already been observed and related with the beginning of the degradation process of the polymer matrices as a result of the water absorption and/or the presence of products generated during the hydrolytic processes [40,51]. The presence of silica aerogel speeds up the disintegration process. The fact that the plasticized PLA based formulations were still visible after 35 days compared to previous works [12,40] is related to the higher thickness of the films developed here as well as to the higher degree of crystallinity promoted by silica aerogel, which could be somewhat enhanced at the temperature of the composting container during the test. An increase in crystallinity of the samples entails a decrease in their degradation rate since the disintegration process starts in the amorphous regions of the polymeric matrix and due to the ordered structure located in the crystalline fractions can retain the microorganisms’ action [52]. Conversely, the presence of ATBC in the formulations leads to a clear increase in the disintegration rate. In fact, at the temperature of the composting test, the plasticized materials are over its T_g_, and thus the presence of a plasticizer increases the polymer chain mobility, facilitating the water diffusion and thus the hydrolysis of polymeric matrix, which consequently further accelerates the disintegration process mediated by the enzymatic attack of microorganisms [8,9,40].

## 4. Conclusions

Silica aerogels were used to reinforce plasticized PLA with ATBC. The incorporation of the silica aerogel in 0.5, 1, 3, and 5 wt.% resulted in a small improvement in the thermal stability of the biocomposites. The presence of a plasticizer decreased, as expected, the glass transition temperature due to the ability of ATBC to increase the mobility of the polymer chains. The resulting higher chain mobility promoted an increment of the degree of crystallinity of PLA as well as the stretchability. On the contrary, it could be observed that the addition of silica aerogel had a clear tendency to decrease the degree of crystallinity with respect to PLA-ATBC samples, which is at the detriment of the enhancement of barrier properties of the composites. Nevertheless, silica aerogel in the amounts up to 3 wt.% was able to generate a more tortuous path which decreased the permeability of the biocomposite films (PLA-ATBC-SiA3% and PLA-ATBC-SiA5%), slightly improving the barrier performance of PLA-ATBC. Although the plasticizer decreased the Young’s modulus and tensile strength, the elongation at break reached values up to 166% in plasticized PLA formulations. In general, no significant modifications were observed in the mechanical behavior of the plasticized biocomposites with the addition of silica aerogel. Nevertheless, in the case of the addition of silica aerogel in 3 wt.%, some improvement of the mechanical resistance was observed which was, however, accompanied with a reduction of the elongation at break, but being stretchable enough (around 140%) for the intended use. PLA-ATBC and its biocomposite films presented a higher disintegration rate than neat PLA due to the presence of the plasticizer, which promote the hydrolysis of polymer chains in the compost medium.

In summary, plasticized PLA loaded with 3 wt.% of silica aerogel formulation, PLA-ATBC-SiA3%, showed the best performance for biodegradable films for food packaging applications with high transparency, improved elongation at break, moderate barrier properties, and being compostable in around 40 days.

## Figures and Tables

**Figure 1 materials-13-04910-f001:**
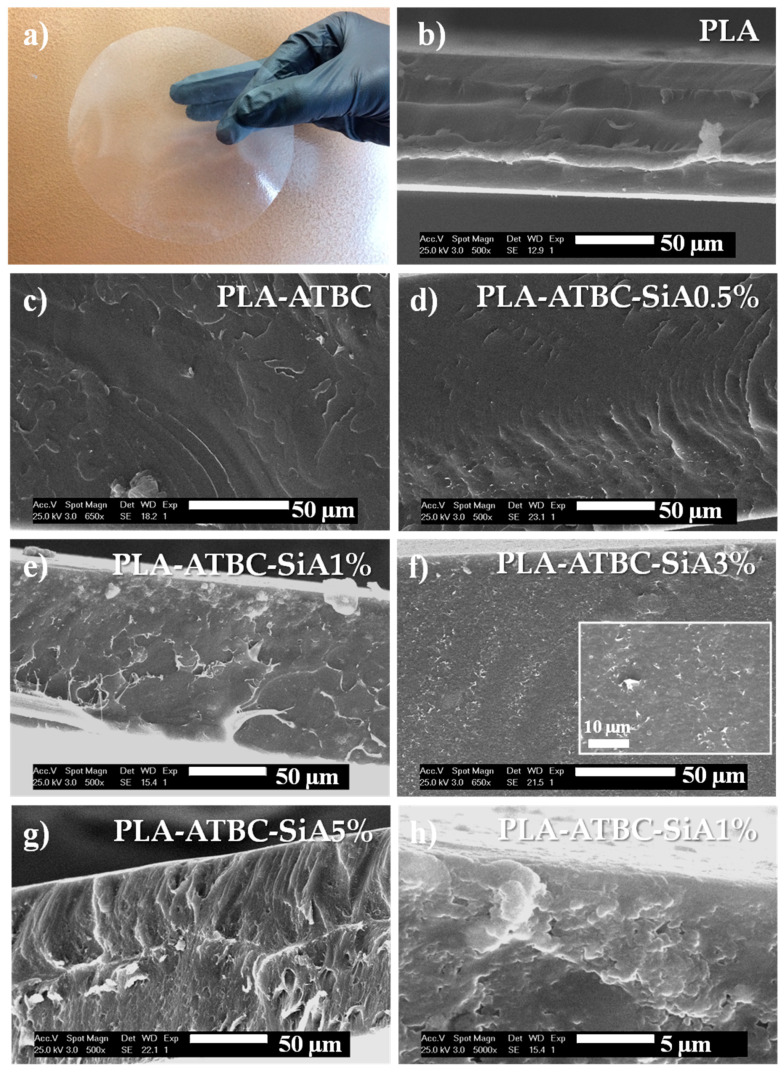
(**a**) Visual appearance of PLA-ATBC-SiA3% composite film. Cryo-fractured SEM micrographs of: (**b**) PLA, (**c**) PLA-ATBC, (**d**) PLA-ATBC-SiA0.5% aerogel, (**e**) PLA-ATBC-SiA1%, (**f**) PLA-ATBC-SiA3% (zoom image at 2000×), (**g**) PLA-ATBC-SiA5%, and (**h**) PLA-ATBC-SiA1% at higher magnification (5000×).

**Figure 2 materials-13-04910-f002:**
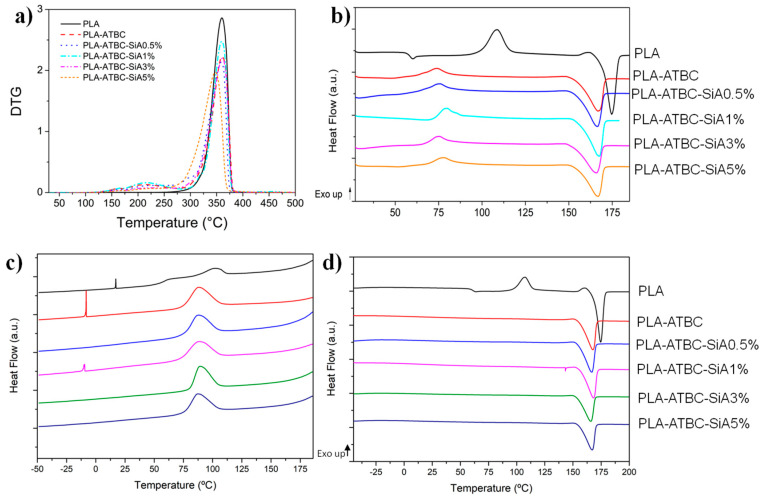
(**a**) DTG curves, (**b**) first heating DSC curves, (**c**) cooling DSC curves, and (**d**) second heating DSC curves of PLA, PLA-ATBC, and hybrid plasticized biocomposite films.

**Figure 3 materials-13-04910-f003:**
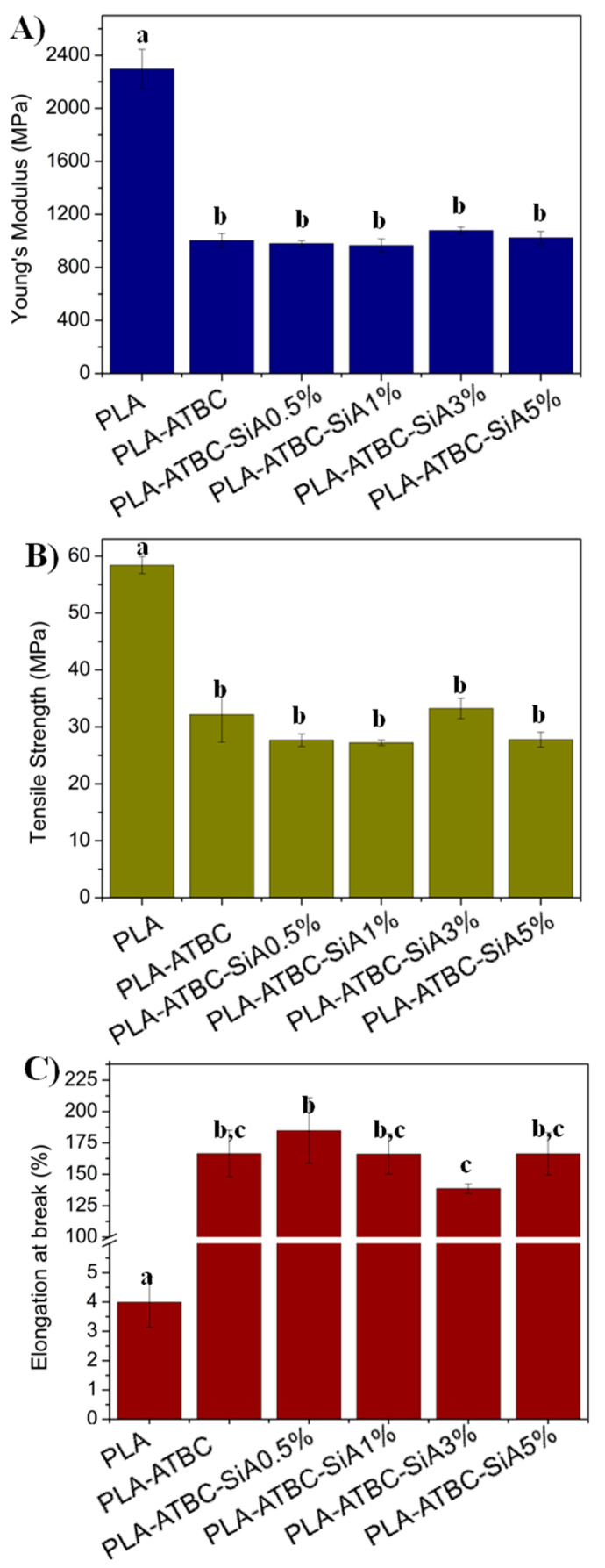
Tensile test results: (**A**) Young Modulus, (**B**) Tensile strength and (**C**) elongation at break. (^a–c^ Different letters on the bars within the same image indicate significant differences between formulations (*p* < 0.05)).

**Figure 4 materials-13-04910-f004:**
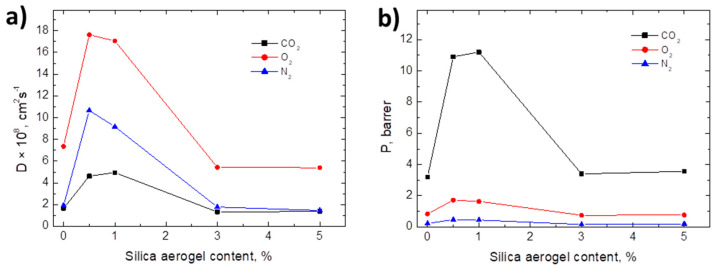
(**a**) Diffusivity and (**b**) permeability as a function of the aerogel content.

**Figure 5 materials-13-04910-f005:**
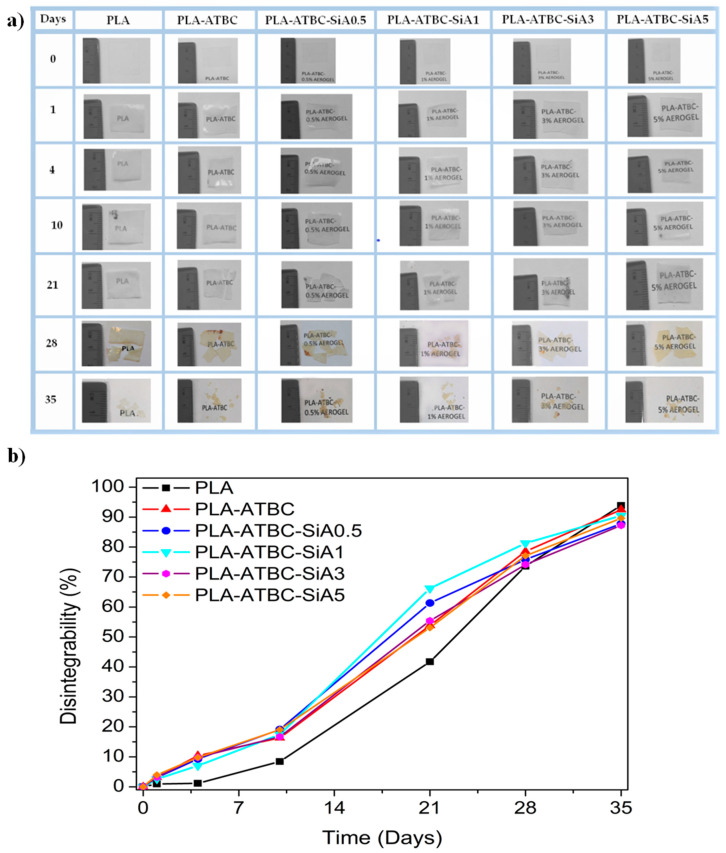
(**a**) visual aspect of film samples at different degradation times, (**b**) degree of disintegration under composting conditions as a function of time.

**Table 1 materials-13-04910-t001:** Biocomposite film formulations.

Formulation	PLA (wt.%)	ATBC (wt.%)	SiA (wt.%)
PLA	100	0	0
PLA-ATBC	85	15	0
PLA-ATBC-SiA0.5%	84.5	15	0.5
PLA-ATBC-SiA1%	84	15	1
PLA-ATBC-SiA3%	82	15	3
PLA-ATBC-SiA5%	80	15	5

**Table 2 materials-13-04910-t002:** TGA and DTG parameters of PLA, PLA-ATBC, and hybrid plasticized biocomposite films.

Film Formulations	T_0_ (°C)	T_max I_ (°C)	T_max II_ (°C)
PLA	322	-	360
PLA-ATBC	205	217	361
PLA-ATBC-SiA0.5%	215	212	358
PLA-ATBC- SiA1%	204	215	360
PLA-ATBC- SiA3%	236	216	361
PLA-ATBC- SiA5%	249	213	348

**Table 3 materials-13-04910-t003:** Thermal parameters of PLA, PLA-ATBC and hybrid plasticized biocomposite films obtained from DSC during the first heating scan.

**Film Formulations**	**T_g_ (°C)**	**T_cc_ (°C)**	**∆H_cc_ (J g^−1^)**	**T_m_ (°C)**		**∆H_m_ (J g^−1^)**		**χ_c_ (%)**
	**First Heating Scan**							
PLA	59	109	30.2	175		52.0		22.1
PLA-ATBC	50	74	13.4	166		54.0		51.3
PLA-ATBC-SiA0.5%	49	75	11.9	166		51.9		50.9
PLA-ATBC-SiA1%	47	79	13.7	167		53.0		50.2
PLA-ATBC-SiA3%	48	75	10.9	166		48.3		49.0
PLA-ATBC-SiA5%	48	78	12.6	167		49.2		49.1
**Film Formulations**	**Cooling Scan**			**Second Heating Scan**				
	**T_g_ (°C)**	**T_cc_ (°C)**	**∆H_cc_ (J g^−1^)**	**T_cc_ (°C)**	**∆H_cc_ (J g^−1^)**	**T_m_ (°C)**	**∆H_m_ (J g^−1^)**	**χ_c_ (%)**
PLA	63	109	21.0	107	4.0	175	53.4	30.5
PLA-ATBC	-	88	28.5	-	0.6	167	46.2	58.4
PLA-ATBC-SiA0.5%	-	88	23.2	-	-	167	46.2	58.7
PLA-ATBC-SiA1%	-	89	24.3	-	-	168	46.8	59.9
PLA-ATBC-SiA3%	-	89	24.3	-	-	166	40.6	53.2
PLA-ATBC-SiA5%	-	87	25.4	-	-	167	40.3	54.2

**Table 4 materials-13-04910-t004:** Comparative values at 30 °C of permeability (P), diffusion (D) and apparent solubility (S) coefficients of carbon dioxide, nitrogen and oxygen in PLA, PLA-ATBC and plasticized biocomposite films.

Film Formulation	Film Thickness (µm)	Gas	P (Barrer)	D × 10^8^ (cm^2^ s^−1^)	S × 10^3^ (cm^3^ (STP) cm^−3^(cm Hg)^−1^)
PLA	99	CO_2_	1.03	0.31	33.1
		N_2_	0.06	0.44	1.3
		O_2_	0.32	1.83	1.7
PLA-ATBC	185	CO_2_	3.21	1.66	19.4
		N_2_	0.22	1.93	1.2
		O_2_	0.84	7.36	1.1
PLA-ATBC-SiA0.5%	185	CO_2_	10.91	4.62	23.6
		N_2_	0.47	10.66	0.4
		O_2_	1.72	17.62	1.0
PLA-ATBC-SiA1%	168	CO_2_	11.21	4.95	22.7
		N_2_	0.44	9.17	0.5
		O_2_	1.64	17.06	1.0
PLA-ATBC-SiA3%	175	CO_2_	3.40	1.33	25.6
		N_2_	0.16	1.78	0.9
		O_2_	0.75	5.42	1.4
PLA-ATBC-SiA5%	197	CO_2_	3.56	1.38	25.9
		N_2_	0.20	1.46	1.4
		O_2_	0.76	5.40	1.4

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
