# Peer review of "Hybrid Biocomposites Based on Poly(Lactic Acid) and Silica Aerogel for Food Packaging Applications"

_materials, 2020, doi:10.3390/ma13214910_

Round 1
Reviewer 1 Report
Materials-983913 - Hybrid biocomposites based on poly(lactic acid) and silica aerogel for food packaging applications
This is a well written study both grammatically and on the technical content. With minor changes, this work can be suggested for publication.
Mention on the risks of having SiA and the plasticizer entering the food chain in the intro.
Line 47. What is meant by “environmentally characteristics”? Environmetally “friendly” charact?
Line 163. Instead of …permeation device was “submitted” look for an appropriate term for example: “subjected to”, “brought to”, “placed under high vacuum”
Line 173. Clarify sentence: Thus, the pressure of permeant against time was represented
Lines 225 – 229. Can you provide quantitative values of how much roughness? What patterns of the figure dictate the level of roughness? Describe the parameter being used to state either more or less roughness.
Lines 225-27. Description of Figures is confusing, clarify. Seems like Figure h is both smooth and rough. “…more ductile fracture patterns (Figure 1-c to h)”, “…resulted more roughness (Figure 1-d to h)”
Line 402, 405 seems to contradict line 439. Please clarify. Typically, the higher the crystallinity (dense regions of aligned molecules) the more strength, makes a material harder, lowering the elongation at break. Therefore, if SiA is increasing crystallinity (line 439), most probably will be lowering the EAB, not making it higher as stated in line 402, unless is decreasing crystallinity as stated in lines 405-06.
Author Response
Materials-983913 - Hybrid biocomposites based on poly(lactic acid) and silica aerogel for food packaging applications
This is a well written study both grammatically and on the technical content. With minor changes, this work can be suggested for publication.
We thanks Reviewer 1 for his/her valuable comments.
Mention on the risks of having SiA and the plasticizer entering the food chain in the intro.
Thank you for this observation, we have introduced some comments in the introduction section regarding the use of citrate esters plasticizers and silica aerogel for food contact applications.
Line 47. What is meant by “environmentally characteristics”? Environmetally “friendly” charact?
Yes. It has been clarified in the current version of the manuscript.
Line 163. Instead of …permeation device was “submitted” look for an appropriate term for example: “subjected to”, “brought to”, “placed under high vacuum”
Thank you for this observation, It has been changed by brought to.
Line 173. Clarify sentence: Thus, the pressure of permeant against time was represented
It has been clarified.
Lines 225 – 229. Can you provide quantitative values of how much roughness? What patterns of the figure dictate the level of roughness? Describe the parameter being used to state either more or less roughness.
Unfortunately, the SEM used cannot quantified the roughness. We have commented the rougher character evidenced from the SEM images. Nevertheless, in the current version of the manuscript, we have introduced some additional comments and also support these statements on the basis of the literature.
Lines 225-27. Description of Figures is confusing, clarify. Seems like Figure h is both smooth and rough. “…more ductile fracture patterns (Figure 1-c to h)”, “…resulted more roughness (Figure 1-d to h)”
The more ductile fracture pattern is due to the presence of plasticizer, while the rougher character is provided by the SiA microparticles.
Line 402, 405 seems to contradict line 439. Please clarify. Typically, the higher the crystallinity (dense regions of aligned molecules) the more strength, makes a material harder, lowering the elongation at break. Therefore, if SiA is increasing crystallinity (line 439), most probably will be lowering the EAB, not making it higher as stated in line 402, unless is decreasing crystallinity as stated in lines 405-06.
In the present work the elongation at break has been improved due to the incorporation of plasticizer. Meanwhile, the introduction of SiA particles into the polymeric system, in general, did not significant changed the mechanical performance of PLA-ATBC. In the current version of the manuscript we have added the ANOVA analysis and better discussed these points.
Reviewer 2 Report
In this manuscript, Silica aerogels (SiA) were used to reinforce plasticized PLA with ATBC. Thermal, mechanical and barrier performance of these hybrid biocomposites were systematically evaluated. However, there exist some drawbacks for these biocomposites in food packaging application. Some questions need to be addressed before recommendation for publication.
- Anti-microbial activity is very important for materials used in food packaging. How about the Anti-microbial activity of hybrid biocomposite?
- Authors mentioned the barrier properties of hybrid biocomposite against carbon dioxide, nitrogen and oxygen, but did not investigate this property for water vapour.
- It is better to perform additional permeation measurements under 4 degree that is relevant to the condition of refrigerator.
- Are these hybrid biocomposites safe for human health? Necessary cell viability assessments should be performed.
Author Response
In this manuscript, Silica aerogels (SiA) were used to reinforce plasticized PLA with ATBC. Thermal, mechanical and barrier performance of these hybrid biocomposites were systematically evaluated. However, there exist some drawbacks for these biocomposites in food packaging application. Some questions need to be addressed before recommendation for publication.
Thank you for your valuable comments.
Anti-microbial activity is very important for materials used in food packaging. How about the Anti-microbial activity of hybrid biocomposite?
The biocomposites developed here are intended for food packaging applications. In the present research the term "bio" is related with the biobased origin and with the biodegradable character of the polymeric matrix in the biocomposites. Although materials with antimicrobial activity are interesting in the food packaging field, the biocomposites developed here do not contain any antimicrobial additive to provide the materials with this property.
Authors mentioned the barrier properties of hybrid biocomposite against carbon dioxide, nitrogen and oxygen, but did not investigate this property for water vapour. It is better to perform additional permeation measurements under 4 degree that is relevant to the condition of refrigerator.
We have investigated the barrier performance of the obtained materials against 3 gases (CO2, NO2 and O2) since it is our field of expertise, while we are not experts in the field of water vapour measurements. Thus, although interesting we do not have the devices required for such measurements. Regarding the permeation measurements at another temperature we think that it is very interesting. However, it represents a high amount of additional permeation measurements (6 samples and 3 gases), that are out of the scope of the present research. These measurements will require to adapt our permeation device to perform permeability measurements at 4 °C and it uses during around 2 months exclusively for this research (which is not possible in this moment). Nevertheless, we have introduce some aspects regarding the permeability performance of PLA at lower temperature on the basis of the literature and we think that it has clarified this point. Moreover, we will consider the possibility to adapt our permeation device to perform permeability measurements at 4 °C for future works.
Are these hybrid biocomposites safe for human health? Necessary cell viability assessments should be performed.
As it was already clarified in response of the point one, the biocomposites developed here are intended for packaging applications and the term "bio" refers to the biobased origin and the biodegradable character of these materials. Thus, considering that cell viability assessments are required for materials intended for biomedical applications, such assessments are out of the scope of the present research.
Reviewer 3 Report
The work of Aragón-Gutierrez et al. deals with the use of SiO2 aerogel as a reinforcing agent for plasticized polylactic acid (PLA) films. The work is well conducted and technically sounding, the results are logically organized and well presented.
Before the publication, I only suggest the preparation of some samples with lower amount of acetyl tributyl citrate (ATBC) used as a plasticizer. Indeed, the authors have chosen to use a PLA containing ATBC in the amount of 15% changing only the SiO2 aerogel amount(0.5-5%). On page 8, lines 287-288 the authors write: "The addition of low amounts 288 of SiA, that is 0.5 wt% and 1 wt%, caused a plasticizing effect." therefore could be also useful to diminish the amount of ATBC in order to find the best match between the two additives.
Author Response
The presented research results and the subject of the article are interesting and deserve publication.
We thanks Reviewer 1 for his/her valuable comments and for considered our manuscript suitable for its publication in Materials journal.
Before publication, the authors should however make some changes and answer a few doubts.
More information about aerogels should be added in the introduction. This is the main element of the work and the literature analysis of this issue is limited to one paragraph only.
Thank you very much for this observation. We have extended the introduction section regarding the use of silica aerogel in the development of polymer composites in the current version of the manuscript.
Was the PLA drying temperature really as high as 80°C? If so, it was definitely too high. Why did the authors dry at such a high temperature?
PLA undergoes hydrolytic and thermal degradation and this is why it have to be dried before processing at this temperature. Thus, it should be dried at least 12 h at 80 °C or 3 h at 98 °C under vacuum. Nevertheless, we have introduced a better explanation of the required drying conditions and we added some additional references of referent research groups in the field of PLA melt-processing.
In my opinion, the authors should analyze the DSC results on the basis of cooling and second heating. Meanwhile, their analysis is based on the first heating cycle, which is decisively influenced by the thermal history of the sample, among others its manufacture, which can make it impossible to accurately determine the effect of fillers on material properties. Please explain the approach used.
Reviewer is totally right. The DSC cooling and second heating scan curves have been added in the current version of the manuscript and they were properly discussed.
Figure 1f - I am not convinced that in the presented magnification we are able to observe aerogel particles that are 20 nm. Therefore, it is difficult to talk about the dispersion assessment.
A zoom image of this SEM image at higher magnification (2000 x) have been added in the current version of the manuscript.
Figure 2 – A DSC cooling curves should also be presented.
The DSC cooling curves have been added in the current version of the manuscript.
The Tg values of samples other than PLA presented in Table 3 are difficult to verify. In the curves presented in Figure 2b, it is difficult to notice glass transitions at the temperatures given by the authors. With such small amplitudes of the glass transition, the authors were sure that the observed curve deviations are the result of the glass transition and not artifacts.
Reviewer is totally right, it is difficult to see the Tg from our DSC curves due to the precision of the equipment. Thus, in the current version of the manuscript we have highlighted the fact that the typical Tg of PLA is not evident at around 60 °C. Moreover, it should be mentioned that the elongation at break considerably increased, thus confirming the ATBC plasticizer effect due to the reduction on Tg values.
Line 358-360 - The observed changes may also result from the increase in the degree of crystallinity of the materials under the influence of the temperature in the composter.
In the current version of the manuscript we have introduce some comments regarding the effect of the temperature in the composter in the study of the materials during composting test.
Reviewer 4 Report
The presented research results and the subject of the article are interesting and deserve publication. Before publication, the authors should however make some changes and answer a few doubts.
- More information about aerogels should be added in the introduction. This is the main element of the work and the literature analysis of this issue is limited to one paragraph only.
- Was the PLA drying temperature really as high as 80°C? If so, it was definitely too high. Why did the authors dry at such a high temperature?
- In my opinion, the authors should analyze the DSC results on the basis of cooling and second heating. Meanwhile, their analysis is based on the first heating cycle, which is decisively influenced by the thermal history of the sample, among others its manufacture, which can make it impossible to accurately determine the effect of fillers on material properties. Please explain the approach used.
- Figure 1f - I am not convinced that in the presented magnification we are able to observe aerogel particles that are 20 nm. Therefore, it is difficult to talk about the dispersion assessment.
- Figure 2 – A DSC cooling curves should also be presented.
- The Tg values of samples other than PLA presented in Table 3 are difficult to verify. In the curves presented in Figure 2b, it is difficult to notice glass transitions at the temperatures given by the authors. With such small amplitudes of the glass transition, the authors were sure that the observed curve deviations are the result of the glass transition and not artifacts.
- Line 358-360 - The observed changes may also result from the increase in the degree of crystallinity of the materials under the influence of the temperature in the composter.
Author Response
The work of Aragón-Gutierrez et al. deals with the use of SiO2 aerogel as a reinforcing agent for plasticized polylactic acid (PLA) films. The work is well conducted and technically sounding, the results are logically organized and well presented.
We thanks Reviewer 1 for his/her valuable comments.
Before the publication, I only suggest the preparation of some samples with lower amount of acetyl tributyl citrate (ATBC) used as a plasticizer. Indeed, the authors have chosen to use a PLA containing ATBC in the amount of 15% changing only the SiO2 aerogel amount(0.5-5%).
The use of ATBC in this amount have been selected on the basis of previous PLA-ATBC based composites and nanocomposites works. Some additional comments have been added in the introduction as well as in the materials and method sections.
On page 8, lines 287-288 the authors write: "The addition of low amounts 288 of SiA, that is 0.5 wt% and 1 wt%, caused a plasticizing effect." therefore could be also useful to diminish the amount of ATBC in order to find the best match between the two additives.
The slightly plasticizing effect produced by the SiA particles is not significant enough to diminish the plasticizer additive in the formulation. Therefore, in the current version of the manuscript we added the statistical ANOVA analysis and we better explained the slightly plasticizing effect provided by the SiA particles.
Round 2
Reviewer 2 Report
Authors have provided good explanation for the questions raised by reviewer previously. The reviewer recommends it for publication.
Author Response
Thankyou very much for accept our work.
Reviewer 4 Report
Although the Authors did not provide written answers to my questions (they pasted the answer to another author by mistake), I can see from the text of the article that they responded to most of my comments. Therefore, I am in favor of accepting the publication.
Author Response
We apologise for that mistake. Thank you very much for considering our work suitable for its publication.